# DemoGrasp: Universal Dexterous Grasping from a Single Demonstration

**Haoqi Yuan**[1,2][*]   **Ziye Huang**[1,2][*]   **Ye Wang**[2,3]   **Chuan Mao**[1]   **Chaoyi Xu**[2]   **Zongqing Lu**[1,2][†]

[1]School of Computer Science, Peking University
[2]BeingBeyond
[3]Renmin University of China

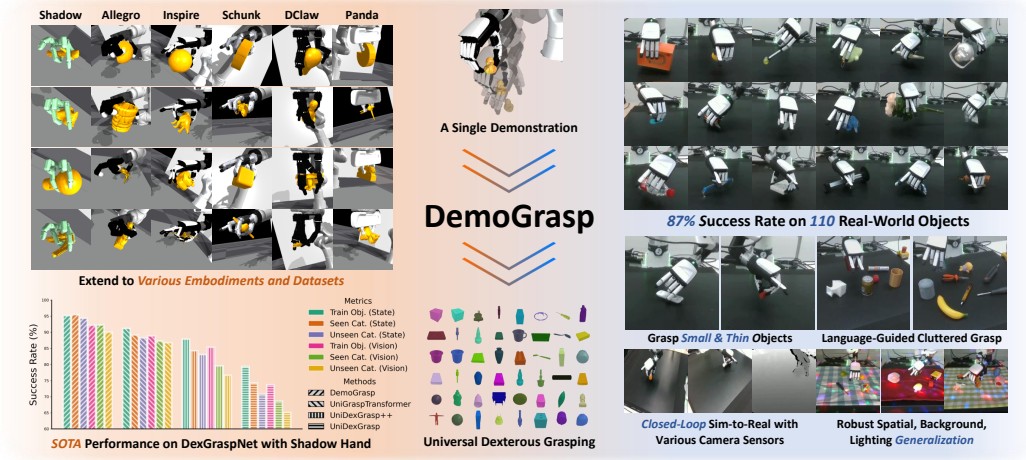

Figure 1: *DemoGrasp* is a framework for learning universal dexterous grasping policies via reinforcement learning (RL) augmented with a single demonstration. It achieves state-of-the-art performance across diverse robotic hand embodiments and transfers effectively to real robots, demonstrating strong generalization.

## Abstract

Universal grasping with multi-fingered dexterous hands is a fundamental challenge in robotic manipulation. While recent approaches successfully learn closed-loop grasping policies using reinforcement learning (RL), the inherent difficulty of high-dimensional, long-horizon exploration necessitates complex reward and curriculum design, often resulting in suboptimal solutions across diverse objects. We propose *DemoGrasp*, a simple yet effective method for learning universal dexterous grasping. We start from a single successful demonstration of grasping a specific object and adapt to novel objects and poses by editing the robot actions in this demonstration: changing the wrist pose determines *where* to grasp, and changing the hand joint angles determines *how* to grasp. We formulate this trajectory editing as a single-step Markov Decision Process (MDP) and use RL to optimize a universal policy across hundreds of objects in parallel in simulation, with a simple reward combining binary success and a robot–table collision penalty. To enable real-world deployment, we collect rollouts using the trained RL policy with rendered images in simulation and apply imitation learning to obtain a closed-loop vision-based policy. In simulation, *DemoGrasp* achieves a 95% success rate on DexGraspNet objects using the Shadow Hand, outperforming previous state-of-the-art methods. It also shows strong transferability, achieving an average success rate of 84.6% across diverse dexterous hand embodiments on six unseen object datasets, while being trained on only 175 objects. In real-world tests, our vision-based policy

---

[*]Equal contribution.
[†]Correspondence: `zongqing.lu@pku.edu.cn`

successfully grasps 110 unseen objects, including small, thin items. It generalizes to spatial, background, and lighting changes, supports both RGB and depth inputs, and extends to language-guided grasping in cluttered scenes. Videos are available on our project page: https://research.beingbeyond.com/demograsp.

# 1 INTRODUCTION

Universal dexterous grasping (Bicchi, 2000; Duan et al., 2021) is a fundamental capability for real-world robots. The anthropomorphic design of dexterous robotic hands makes them the most suitable manipulators for real-world manipulation tasks, such as tool use, in-hand reorientation, and bimanual coordination. Universal grasping is therefore an essential prerequisite for enabling these sophisticated interactions. Though basic in concept, learning universal dexterous grasping policies remains far from simple. The high-dimensional action space introduced by dexterous hands with many degrees of freedom (DoFs), together with the long-horizon nature of closed-loop grasping, imposes substantial exploration challenges for reinforcement learning (RL). At the same time, the diverse geometries of objects make universal dexterous grasping a multi-task optimization problem, introducing additional difficulties such as catastrophic forgetting (Kirkpatrick et al., 2017; Schwarz et al., 2018) and gradient interference (Teh et al., 2017; Yu et al., 2020).

Recent studies have extensively investigated the use of RL for training universal dexterous grasping policies. Xu et al. (2023); Wan et al. (2023); Zhang et al. (2025b); Chen et al. (2025b) introduce techniques in observation feature design, dense reward shaping, and curriculum learning strategies to facilitate policy learning. UniDexGrasp++ (Wan et al., 2023) employs an iterative distillation process to improve teacher–student learning. ResDex (Huang et al., 2025) introduces a two-stage residual RL framework to accelerate multi-task exploration. UniGraspTransformer (Wang et al., 2025) proposes exhaustive RL on individual objects and distillation with expressive Transformer policies to bypass multi-task RL. However, many of these approaches train on hands without robot arms (Xu et al., 2023; Wan et al., 2023), use privileged contact information as observations (Wan et al., 2023; Huang et al., 2025), and face a trade-off between collision penalties and other complex reward terms (Xu et al., 2023; Huang et al., 2025), limiting their potential for deployment on real robots. Singh et al. (2024); Zhang et al. (2025b) achieve sim-to-real on a wide variety of objects but still fall short on grasping small, thin objects in tabletop settings. In addition, their reliance on complicated observation design, reward shaping, and multi-stage pipelines increases the barrier to extending these methods to new embodiments and task settings.

In this research, we propose *DemoGrasp*, a simple yet powerful framework for universal dexterous grasping that addresses these challenges. Our key insight is that a single demonstration trajectory of grasping a specific object encodes many transferable patterns for universal grasping, such as approaching the object's grasp center, squeezing the hand pose, and lifting the wrist. To grasp various objects in different poses, we can slightly modify the robot actions within this trajectory and replay the edited actions. For example, to grasp the same object at a different location, we can apply a transformation to the wrist poses in the trajectory, changing *where to grasp*; to grasp a larger object at the same position, we adjust the grasp poses to be more open, changing *how to grasp*. In our method, the RL policy explores how to edit the demonstration along these two axes, rather than exploring in the low-level robot action space as in prior methods (Xu et al., 2023; Wang et al., 2025; Zhang et al., 2025b), resulting in more efficient trial-and-error.

Specifically, we formulate the demonstration-editing task as a single-step Markov Decision Process (MDP). At each trial, given an arbitrary object placed at a random position, the policy outputs an $SE(3)$ transformation and delta hand joint angles, which are used to modify the end-effector poses and hand actions in the demonstration. The edited demonstration is then replayed in simulation, yielding a reward for the whole episode. By restricting the policy to a compact action space and a single-step decision-making horizon, the multi-task exploration burden is significantly reduced, removing the need for complex reward shaping. This enables us to effectively train a universal grasping policy on hundreds or thousands of objects by optimizing a simple combination of binary success reward and a collision penalty. We observe that this design yields both superior performance in simulation and easy sim-to-real transfer with minimal collisions. We train a flow-matching (Lipman et al., 2022) policy on successful rollouts of the learned policy with rendered camera images in simulation, enabling zero-shot deployment on a real robot. Figure 2 provides an overview of our approach.

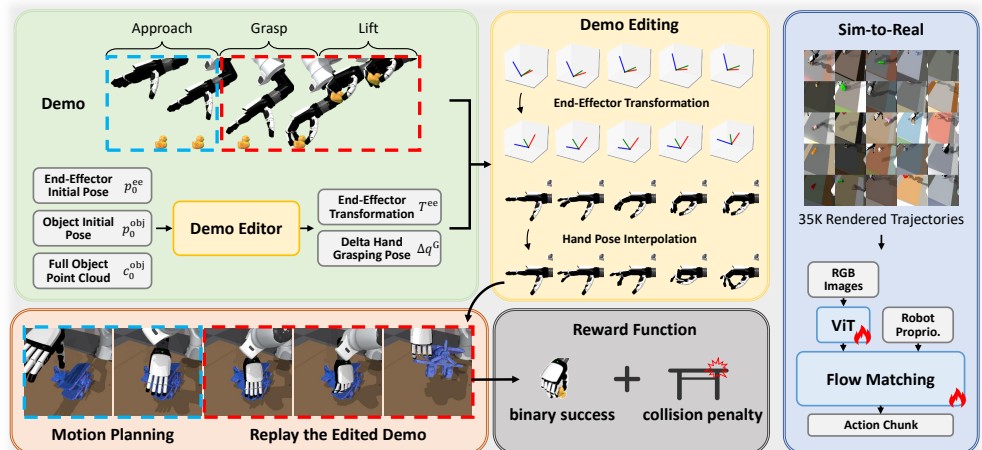

Figure 2: *DemoGrasp* uses a single demonstration trajectory to learn universal dexterous grasping, formulating each grasping trial as a demonstration-editing process. For each trial, the Demo Editor policy takes observations at the first timestep and outputs an end-effector transformation and a delta hand pose. The actions in the demonstration are then transformed accordingly and applied in the simulator. The policy is trained using RL across diverse objects, optimizing a simple reward consisting of binary success and a collision penalty. A flow-matching policy is trained on successful rollouts with rendered images to enable sim-to-real transfer.

We conduct large-scale experiments in both simulation and the real world to evaluate *DemoGrasp*. On 3.4K objects from DexGraspNet (Wang et al., 2023), *DemoGrasp* achieves success rates of 95% in state-based settings and 92% in vision-based settings, surpassing previous state-of-the-art method (Wang et al., 2025) by a large margin. *DemoGrasp* also exhibits strong transferability to a wide variety of robotic embodiments and generalization to unseen object categories. Trained on 175 objects, the policies achieve an average success rate of 84.6% on six unseen object datasets across various embodiments, including dexterous hands with different numbers of fingers, grippers, and arm–hand systems. In real-world experiments, *DemoGrasp* achieves a success rate of 86.5% on 110 unseen objects, covering a wide variety of geometries and visual appearances. For normal-sized objects, it achieves a superior success rate of 95.3%. Benefiting from the simple reward design, the policy is, to our knowledge, the first to grasp previously unseen small, thin objects in tabletop settings without severe collisions, achieving a success rate of 71.1%. *DemoGrasp* also exhibits generalization to spatial, background, and lighting changes, and is extensible to various camera configurations (RGB and depth) and cluttered scenes, underscoring its practical applicability.

Our contributions are summarized as follows:

- We propose *DemoGrasp*, a simple yet powerful learning framework that addresses key challenges in learning universal dexterous grasping policies. With a novel formulation of demonstration editing and single-step RL, *DemoGrasp* enables robust policy learning, minimal reliance on reward shaping, and sim-to-real transferability.

- *DemoGrasp* achieves state-of-the-art performance in large-scale evaluations in both simulation and the real world, demonstrating strong capability in grasping unseen objects.

- We demonstrate the strong extensibility of *DemoGrasp* to novel embodiments, camera configurations, and cluttered scenes, establishing a foundation for future research and applications in dexterous manipulation.

## 2 METHOD

### 2.1 PROBLEM FORMULATION

We consider grasping an arbitrary object from a large object set in tabletop settings. The task is formulated as a partially observable Markov Decision Process (MDP) (Kaelbling et al., 1998).

Specifically, at each timestep $t$, the observation comprises the hand joint angles $q_t^{\mathrm{hand}}$, the 6D pose of the end-effector (wrist) $p_t^{\mathrm{ee}}$, the 6D pose of the object $p_t^{\mathrm{obj}}$, and a full object point cloud $c_t^{\mathrm{obj}}$ describing its geometry; the action comprises target hand joint angles $\hat{q}_t^{\mathrm{hand}}$ and the target end-effector pose $\hat{p}_t^{\mathrm{ee}}$ for the PD controller. The objective is to learn a universal state-based policy

$$\pi\left(\hat{q}_t^{\mathrm{hand}}, \hat{p}_t^{\mathrm{ee}} \,\middle|\, q_t^{\mathrm{hand}}, p_t^{\mathrm{ee}}, p_t^{\mathrm{obj}}, c_t^{\mathrm{obj}}\right)$$

that maximizes the expected cumulative reward $\mathbb{E}\left[\sum_{t=0}^{T-1} \gamma^t r_t\right]$ across objects, where $r_t$ denotes a task reward encouraging successful grasping, $T$ is the time limit, and $\gamma$ is the discount factor. Since solving this multi-step MDP directly is challenging, we present our techniques in the following subsections, with the final formulation as a single-step MDP in Section 2.3.

To enable sim-to-real transfer — where object poses and full object point clouds are not observable on hardware — we follow common approaches (Singh et al., 2024; Zhang et al., 2025b) that train a vision-based policy

$$\pi^{\mathrm{vision}}\left(\hat{q}_t^{\mathrm{hand}}, \hat{p}_t^{\mathrm{ee}} \,\middle|\, q_t^{\mathrm{hand}}, p_t^{\mathrm{ee}}, v_t\right)$$

to imitate the learned state-based policy, where $v_t$ denotes visual input (e.g., RGB images, depth images, or partial point clouds).

## 2.2 DEMONSTRATION EDITING

The high-dimensional actions, long task horizons, and multi-task nature of training on all objects in the formulated MDP pose significant challenges for RL exploration. We propose using a single demonstration to facilitate exploration.

The demonstration is a trajectory of a successful grasp for a specific object in the simulator, which can be acquired either by teleoperation in the simulator or by executing a hard-coded robot action sequence. We define the *initial object frame* as a static coordinate frame obtained by translating the world frame to the object's geometric center at the first timestep of the demonstration. We then represent the robot actions in the demonstration in this frame:

$$D = \{(q_t^{*\mathrm{hand}}, \, p_t^{*\mathrm{ee\text{-}obj}})\}_{t=0}^{T^D},$$

where $q_t^{*\mathrm{hand}}$ denotes the target hand joint angles and $p_t^{*\mathrm{ee\text{-}obj}}$ denotes the target 6D end-effector pose expressed in the initial object frame. Intuitively, $\{q_t^{*\mathrm{hand}}\}$ forms a hand-pose sequence from open to close, and $\{p_t^{*\mathrm{ee\text{-}obj}}\}$ is an end-effector trajectory that first approaches the object center (the origin of the initial object frame) and then lifts.

For an arbitrary object placed at any position in the simulator, we can attempt to grasp it by simply *replaying* this demonstration in an open-loop manner: (1) first, set the hand action to $q_0^{*\mathrm{hand}}$ and move the end effector to $p_0^{*\mathrm{ee\text{-}obj}}$ under the new initial object frame via motion planning, aligning the robot's pose (in that frame) with the first step of the demonstration; (2) then, for each timestep $t = 1, \cdots, T^D$, set the hand action to $q_t^{*\mathrm{hand}}$ and transform $p_t^{*\mathrm{ee\text{-}obj}}$ back to the world frame as the end-effector's target pose in the world frame. This replay mechanism, which reuses the same hand grasp poses and wrist approach directions for all objects, already achieves non-trivial success rates when evaluated across all objects (see Table 8).

Universal grasping necessitates more flexibility in motion patterns than replaying a single predefined trajectory. For example, for large objects with different graspable parts, or thin, slippery objects that require the fingers to reach under the object, the end-effector pose sequence $\left\{p_t^{*\mathrm{ee\text{-}obj}}\right\}$ should be adjusted to change *where to grasp*. For objects with varied sizes and geometric features, the hand action sequence $\left\{q_t^{*\mathrm{hand}}\right\}$ should be adjusted to change *how to grasp*. Hence, we introduce parameters for demonstration editing to adapt the demonstration to diverse objects. The parameters consist of an end-effector transformation matrix $T^{\mathrm{ee}} \in \mathrm{SE}(3)$ and delta joint angles for the hand

$\Delta q^{\mathrm{G}}$. Robot actions in the demonstration are then modified as:

$$p_t^{*'\text{ee-obj}} = \begin{cases} T^{\text{ee}} \, p_t^{*\text{ee-obj}}, & t \leq T_{\text{lift}}, \\ \begin{bmatrix} \boldsymbol{I} & \Delta \boldsymbol{z} \\ \boldsymbol{0} & 1 \end{bmatrix} p_{T_{\text{lift}}}^{*'\text{ee-obj}}, & \text{otherwise}, \end{cases} \tag{1}$$

$$q_t^{*'\text{hand}} = \begin{cases} q_0^{*\text{hand}} + (q_t^{*\text{hand}} - q_0^{*\text{hand}}) \left( \dfrac{q_{T_{\text{lift}}}^{*\text{hand}} + \Delta q^{\mathrm{G}} - q_0^{*\text{hand}}}{q_{T_{\text{lift}}}^{*\text{hand}} - q_0^{*\text{hand}}} \right), & t \leq T_{\text{lift}}, \\ q_{T_{\text{lift}}}^{*'\text{hand}}, & \text{otherwise}. \end{cases} \tag{2}$$

Here, $T_{\text{lift}}$ denotes the first timestep at which the object's $z$-position increases (i.e., it begins to be lifted) in the demonstration. $\Delta \boldsymbol{z}$ is a constant vector in the $z$ direction that lifts the object vertically after $T_{\text{lift}}$. End-effector target poses are modified by applying the transformation $T^{\text{ee}}$ in the initial object frame, changing the approach direction and offset toward the object center. Hand actions at each timestep are modified by interpolating between the initial open pose $q_0^{*\text{hand}}$ and the modified grasp pose $q_{T_{\text{lift}}}^{*\text{hand}} + \Delta q^{\mathrm{G}}$; the interpolation ratio is applied elementwise.

We denote the edited demonstration as $D' = \text{Edit}(D, T^{\text{ee}}, \Delta q^{\mathrm{G}})$. ***By varying $T^{\text{ee}}$ and $\Delta q^{\mathrm{G}}$ and replaying $D'$ in simulation, the robot executes diverse, smooth action sequences that grasp the object at different positions, orientations, and hand poses, yielding an effective exploration scheme for universal grasping.***

### 2.3 SINGLE-STEP REINFORCEMENT LEARNING

**MDP reformulation.** Given the grasp exploration scheme via demonstration editing, we reformulate the task as a **single-step MDP**: the policy outputs a single action specifying the editing parameters, after which the edited demonstration is replayed in the environment for a maximum of $T$ timesteps, and the environment returns a reward for the whole episode. Formally, the observation comprises the initial end-effector 6D pose $p_0^{\text{ee}}$, the initial object pose $p_0^{\text{obj}}$, and the full object point cloud $c_0^{\text{obj}}$. The action consists of the end-effector transformation $T^{\text{ee}}$ and the delta hand grasp pose $\Delta q^{\mathrm{G}}$ used for demonstration editing. The transition replays the edited demonstration $D'$ and then terminates. The policy $\pi(T^{\text{ee}}, \Delta q^{\mathrm{G}} \mid p_0^{\text{ee}}, p_0^{\text{obj}}, c_0^{\text{obj}})$ aims to maximize the expected single-step reward $\mathbb{E}[r]$. In implementation, we represent end-effector rotations as quaternions in the observation space and as Euler angles in the action space, yielding a compact representation.

**Reward design.** With the compact, low-dimensional action space and the short horizon introduced by the one-step MDP, the exploration challenge is significantly mitigated, making complicated reward engineering unnecessary. We therefore use a simple reward that comprises grasp success and robot–table collisions, focusing the policy on collision-free grasping:

$$r = \mathbf{1}[\text{success}] \cdot \mathbf{1}[\text{no collision during execution}]. \tag{3}$$

However, grasping flat objects on the table sometimes requires slight contact with the surface so that the fingers can reach underneath the object. The strict collision-free objective may prevent success in these cases. To address this, we leverage IsaacGym's parallel simulation (Makoviychuk et al., 2021) to optimize across all objects simultaneously and randomly disable robot–table collision detection in half of the environments, allowing hand–table penetration. In the reward, collisions are assessed via penetration of hand keypoints into the table. This design yields: (1) collision-free successful grasps achieve $\mathbb{E}[r] = 1$; (2) successful grasps with robot–table contact receive $\mathbb{E}[r] = 0.5$; and (3) failures receive $\mathbb{E}[r] = 0$. This encourages the policy to avoid unnecessary collisions while permitting minimal contact when beneficial for hard-to-grasp objects.

### 2.4 VISION-BASED SIM-TO-REAL

After training the RL policy, we train a vision-based policy on its successful rollouts to enable sim-to-real transfer. We record robot proprioception (hand joint angles and end-effector poses), robot actions, and rendered RGB or depth images from successful rollouts to form a dataset. We then train a Flow-Matching (Lipman et al., 2022) policy with action chunking for imitation learning, modeling the multi-modal action distribution with high quality. To close the visual sim-to-real gap, we perform domain randomization of colors, textures, lighting conditions, camera extrinsics, and table positions

Table 1: **Success rates on DexGraspNet with the Shadow Hand in simulation.** Results are reported for both state-based and vision-based settings on 3,200 training objects (Train.), 141 unseen objects from seen categories (Test Seen Cat.), and 100 unseen objects from unseen categories (Test Unseen Cat.).

| Method | State-Based Setting (%) | | | Vision-Based Setting (%) | | |
|---|---|---|---|---|---|---|
| | Train. | Test. Seen Cat. | Test. Unseen Cat. | Train. | Test. Seen Cat. | Test. Unseen Cat. |
| UniDexGrasp | 79.4 | 74.3 | 70.8 | 73.7 | 68.6 | 65.1 |
| UniDexGrasp++ | 87.9 | 84.3 | 83.1 | 85.4 | 79.6 | 76.7 |
| UniGraspTransformer | 91.2 | 89.2 | 88.3 | 88.9 | 87.3 | 86.8 |
| DemoGrasp | **95.2** | **95.5** | **94.4** | **92.2** | **92.3** | **90.1** |

during data collection, and we finetune a pre-trained ViT (Dosovitskiy et al., 2021) encoder for the visual representation. Further implementation details are provided in Appendix E.

## 3 EXPERIMENTS

Our experiments aim to evaluate: (1) the performance and scalability of our method through large-scale simulation with diverse object datasets and dexterous hand embodiments (Sections 3.2 and 3.3); (2) the sim-to-real performance of our method through real-world experiments with a wide variety of objects (Section 3.4); and (3) an analysis of the components of the proposed method (Section 3.5).

### 3.1 EXPERIMENTAL SETTINGS

**Simulation.** We use IsaacGym (Makoviychuk et al., 2021) as the training and evaluation platform for all simulation experiments. For evaluations on DexGraspNet (Wang et al., 2023) with the Shadow Hand (Section 3.2), we train on 3,200 objects from the DexGraspNet training set to align with baseline settings. In all remaining sections, unless otherwise specified, we randomly sample 175 objects from the YCB dataset (Calli et al., 2015) and the DexGraspNet training set for training, and test on unseen objects from other datasets. For both training and evaluation, we randomize the object's initial position within a $50\,\mathrm{cm} \times 50\,\mathrm{cm}$ region to ensure spatial generalization of the policy. A trial is considered successful if the object's center is raised at least $10\,\mathrm{cm}$ above its original position and the average distance between the object's center and hand keypoints is less than $12\,\mathrm{cm}$ after the policy executes for a fixed number of steps.

**Real robot.** We use a 6-DoF Inspire Hand (6 active and 6 passive joints) mounted on a 7-DoF Franka Research 3 (FR3) robot arm for real-world experiments. Two RealSense D435i cameras are used to evaluate vision-based policies with either RGB or depth input, placed at two diagonal sides of the table. In simulation, the camera intrinsics match those of the real cameras, and the camera extrinsics are randomized around the calibrated real-camera extrinsics. Figure 12 shows the hardware setup and the camera views.

### 3.2 RESULTS ON DEXGRASPNET

DexGraspNet (Wang et al., 2023) is a widely used dataset for studying universal dexterous grasping. We follow the settings of previous state-of-the-art methods—UniDexGrasp (Xu et al., 2023), UniDexGrasp++ (Wan et al., 2023), and UniGraspTransformer (Wang et al., 2025)—training the 18-DoF Shadow Hand with a 6-DoF floating wrist on the training set of 3,200 objects. Table 1 reports grasp success rates for *DemoGrasp* and prior methods. *DemoGrasp* surpasses the best baseline by 5% in state-based settings and 4% in vision-based settings on both training and test sets, and exhibits a minimal generalization gap of 1% between training and unseen objects, demonstrating strong learning and generalization performance.

Notably, the baseline methods do not randomize object initial positions, whereas our method is trained and tested with a large reset region of $50\,\mathrm{cm} \times 50\,\mathrm{cm}$, posing a challenge for spatial generalization. Benefiting from the translation invariance of our demonstration-replay mechanism (i.e., replaying a

Table 2: **Success rates across unseen datasets in simulation using the Allegro Hand mounted on a UR5 arm.**

| Method \ Dataset | DGA | EGAD | Omni6DPose | ModelNet40 | VisualDexterity |
|---|---|---|---|---|---|
| RobustDexGrasp | 64.40 | 93.45 | 73.00 | **75.70** | 92.50 |
| DemoGrasp | **74.40** | **96.75** | **82.24** | 75.58 | **97.80** |

demonstration for the same object at different initial locations leads to the same grasp outcome), spatial randomization does not hinder RL exploration in our method, yielding strong spatial generalization. In addition, while baselines rely on complex reward designs (e.g., hand–object distance, object-lift, and hand-lift terms) to facilitate RL, our method uses a simple binary reward, highlighting the simplicity and effectiveness of our approach.

### 3.3 SCALABILITY AND GENERALIZATION

Previous research on tabletop dexterous grasping has typically evaluated a narrow set of datasets and a specific dexterous hand, leaving method scalability largely unassessed. Zhong et al. (2025b) find that existing datasets, such as DexGraspNet, do not span the breadth of real-world graspable objects. Therefore, we conduct cross-dataset zero-shot tests to evaluate the generalization of universal grasping policies. We train the policy on 175 objects—75 randomly sampled from YCB (Calli et al., 2015) and 100 randomly sampled from the DexGraspNet training set (Wang et al., 2023)—and test on five out-of-distribution datasets: DGA (Zhong et al., 2025b), EGAD (Morrison et al., 2020), Omni6DPose (Zhang et al., 2024b), ModelNet40 (Wu et al., 2015), and Visual Dexterity (Chen et al., 2023). For Omni6DPose and ModelNet40, which consist of larger objects, we randomly scale objects to sizes between $6\,\mathrm{cm}$ and $15\,\mathrm{cm}$ for testing. A snapshot of objects from each dataset is provided in Figure 6.

We evaluate *DemoGrasp* on various robotic hands without hyperparameter tuning, assessing its cross-embodiment universality. For the Allegro Hand mounted on a UR5 arm, we compare against the RobustDexGrasp (Zhang et al., 2025b) policy. Although trained on different object datasets, the test sets are unseen for both methods and thus form a fair comparison, since both aim at universal grasping over arbitrary objects. As shown in Table 2, *DemoGrasp* matches RobustDexGrasp on ModelNet40 and surpasses it on the other four datasets, demonstrating the strong generalizability of the *DemoGrasp* policy.

We further extend the evaluation to various embodiments from Ding et al. (2024), including five-fingered hands (Inspire Hand, Shadow Hand, and Schunk SVH Hand), the four-fingered Allegro Hand, the three-fingered DClaw gripper, and a parallel gripper. Figure 3 visualizes results for all hands on the test datasets, and the quantitative results are reported in Table 10. All multi-fingered hands achieve $> 90\%$ success on the 175 training objects and generalize to unseen datasets with an average success rate of $84.6\%$, indicating that our method extends easily to different hands rather than overfitting to a particular hand. Notably, all six hands are mounted on robot arms;

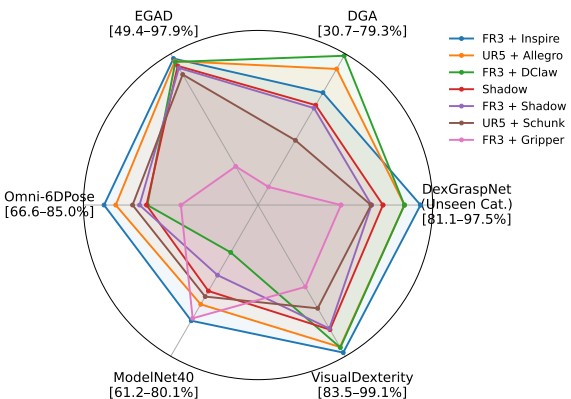

Figure 3: **Success rates of *DemoGrasp* for various robotic embodiments across all test datasets.**

together with the collision-free training objective, this makes the trained policies more likely to succeed in sim-to-real deployment compared with prior work using floating-wrist hands. Our results show that the Shadow Hand mounted on an arm (FR3+Shadow) underperforms the floating Shadow Hand (Shadow) by only $1.4\%$ on average across the test sets, indicating that adding a robot arm does not harm performance. FR3+Gripper performs worse on EGAD and DGA, because the Panda gripper's limited stroke hinders grasping wide or large objects; in contrast, all multi-fingered hands achieve high success rates on EGAD.

## 3.4 REAL-WORLD EXPERIMENTS

We evaluate the vision-based policies on a real robot using 110 unseen real-world objects to assess generalization and sim-to-real performance. Images of all objects are shown in Figure 5. For each trial, we randomize the initial object orientations and positions within a $50\,\text{cm} \times 50\,\text{cm}$ region and count success when the object is lifted and held for two seconds. Table 3 reports the success rates of *DemoGrasp* using two RGB views across different categories of real-world objects. It achieves an average success rate of 95.3% on normal-sized objects—including everyday items of various shapes, deformable objects, and irregular geometries—demonstrating strong generalization. For flat, thin objects (thickness $< 1.5\,\text{cm}$), it achieves 68.3%; for small objects (diameter $< 3.5\,\text{cm}$), it achieves 76.7%, demonstrating successful sim-to-real on these challenging tabletop grasping scenarios. Figure 7 illustrates trajectories in real-world tests, showing that *DemoGrasp* can achieve collision-free grasps for normal-sized objects, appropriately leverage finger–table contact to grasp small, thin objects, and exhibit regrasp behaviors to recover from failures in a closed-loop manner.

*DemoGrasp* is also extensible to more advanced grasping tasks, such as cluttered grasping and instruction-following, by including random distractor objects and automatically generated language descriptions during vision-based data collection in simulation. We evaluate the policies in both simulation and the real world. In simulation, we test on randomly sampled cluttered scenes; in the real world, we test on 10 cluttered scenes, each consisting of 5–8 randomly selected objects. Table 4 reports their success rates. The unconditional policy (Any-*DemoGrasp*) counts success when it grasps any object from the clutter, whereas the language-conditioned policy (Instruct-*DemoGrasp*) counts success when it grasps the object specified by the instruction. Both policies achieve $>80\%$ success in both simulation and the real world, highlighting the robust performance of *DemoGrasp* on challenging task settings. The language-conditioned policy slightly outperforms the unconditional policy, suggesting that reducing action uncertainty for vision-based imitation learning can improve action quality. We further evaluate the language-conditioned policy under randomized backgrounds and lighting conditions, achieving an 82% success rate, demonstrating robustness to scene appearance changes. Qualitative results are provided in Appendix D.2.

Table 3: **Success rates on 110 real-world objects.** Each object is tested for five trials with randomized initial poses.

| Shape | Category | Num. | Success. |
|---|---|---|---|
| Regular | Bottles | 12 | 95.0% |
| | Boxes & Jars | 22 | 93.6% |
| | Balls & Fruit | 12 | 98.3% |
| | Soft Toys | 10 | 96.0% |
| Irregular | | 18 | 90.0% |
| Flat & Thin | Tools | 10 | 60.0% |
| | Others | 14 | 74.3% |
| Small | | 12 | 76.7% |

Table 4: **Results for grasping in cluttered scenes.** "Any-*DemoGrasp*" denotes the unconditioned policy that grasps a random object; "Instruct-*DemoGrasp*" denotes the language-conditioned policy.

| Model | Sim. | Real. |
|---|---|---|
| **Any-*DemoGrasp*** | 83.66% | 82% |
| **Instruct-*DemoGrasp*** | 85.33% | 84% |

## 3.5 ABLATION STUDY

**The necessity of RL.** A direct ablation for *DemoGrasp* is to replace RL with sampling-based methods. Given the demonstration-editing scheme, we sample in the editing-parameter space for each object and position, execute the edited rollout in simulation, and train a behavior cloning (BC) policy on the successful rollouts. Table 5 compares this sampling-based method with the RL-based method on the 175 training objects. We find that sampling+BC policies achieve significantly lower success rates than the RL policy. The main reason is that sampling can produce diverse successful rollouts for the same object and position, yielding a multimodal and inconsistent dataset that hinders BC from converging to an

Table 5: **Sampling vs. RL.** For the sampling-based method, we uniformly sample editing parameters to collect 35,000 successful trajectories, then train a behavior cloning (BC) policy.

| Method | Success (%) |
|---|---|
| Sampling | 77.56 |
| RL | **96.24** |

Table 6: **Success rates of vision-based policies with different camera configurations in simulation and the real world.**

| Camera Config. | Simulator | | Real Robot | | | | |
|---|---|---|---|---|---|---|---|
| | YCB | DexGraspNet | black bottle | blue box | little duck | tiny bottle | phone case |
| Mono-Depth | 80.2% | 95.2% | **5**/5 | 4/5 | 1/5 | 0/5 | 0/5 |
| Two-Depth | 80.3% | 96.4% | **5**/5 | 4/5 | 1/5 | 0/5 | 0/5 |
| Mono-RGB | 83.2% | 94.8% | 4/5 | **5**/5 | **4**/5 | 0/5 | 3/5 |
| Two-RGB | **87.0%** | **97.3%** | **5**/5 | **5**/5 | **4**/5 | **5**/5 | **5**/5 |

Table 7: **Success rates on the test sets** when trained on 175 objects from the training set (row 1) or trained directly on the union of the test sets (row 2).

| Test \ Train | DGA | EGAD | Omni-6DPose | ModelNet40 | Visual Dexterity |
|---|---|---|---|---|---|
| 175 Objects (YCB+DexGraspNet) | 65.62 | 97.88 | 85.04 | 80.13 | 99.13 |
| Test Sets | 71.49 | 99.16 | 88.71 | 81.10 | 99.20 |

optimal policy. In contrast, RL directly optimizes the expected return, resulting in a more consistent, unimodal policy with higher success rates.

**The action space in RL.** We examine the effect of each demonstration-editing parameter on RL performance, studying the contributions of wrist DoFs and hand DoFs. Table 8 reports success rates for RL with different action-space components. Success rates consistently improve as the action space expands, showing that RL can effectively leverage the full dexterity of the wrist and hand to achieve higher performance. Editing end-effector translations and rotations is essential for grasping, contributing +6% and +13% to the training-set success rate, respectively; editing hand DoFs yields a smaller +2% gain, indicating that using dexterous hands as single-DoF grippers can already achieve high success rates in grasping. Figure 4 further shows that editing hand DoFs produces more robust grasps (e.g., grasping a vase from the side and using the thumb, index, and ring fingers to form force closure), whereas other ablations do not exhibit this behavior.

Table 8: **Success rates of RL policies with different action spaces.** $\Delta$xyz, $\Delta$rpy, and $\Delta\mathbf{q}$ denote the inclusion of wrist translation, wrist rotation, and delta hand actions in the RL action space, respectively. The first row corresponds to replaying the original demonstration without RL.

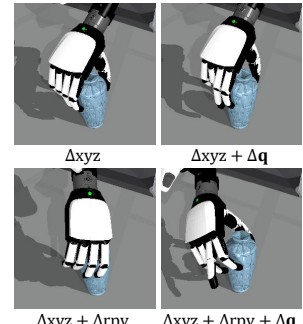

$\Delta$xyz    $\Delta$xyz + $\Delta\mathbf{q}$

$\Delta$xyz + $\Delta$rpy    $\Delta$xyz + $\Delta$rpy + $\Delta\mathbf{q}$

Figure 4: **Learned grasps under different action spaces.**

| $\Delta$xyz | $\Delta$rpy | $\Delta\mathbf{q}$ | Training Set | Test Set |
|---|---|---|---|---|
| | | | 75.29 | 73.43 |
| ✓ | | | 81.35 | 76.04 |
| ✓ | | ✓ | 86.40 | 79.68 |
| ✓ | ✓ | | 94.22 | 81.39 |
| ✓ | ✓ | ✓ | **96.24** | **82.74** |

**Camera configurations for vision-based policies.** *DemoGrasp* enables sim-to-real deployment with different camera types by training vision-based policies on correspondingly rendered data. Table 6 reports performance using either RGB or depth input, and either monocular or two-view configurations. We observe that two RGB views achieve the best performance in both simulation and the real world, outperforming the two-depth-camera configuration. Specifically, RGB policies achieve >70% success rates on small, flat objects in the real world, whereas depth policies often fail. This is because RGB policies can identify such objects via visual cues (e.g., color and texture), whereas depth policies may not distinguish the object from the tabletop due to sensor noise. Two-camera setups consistently outperform monocular ones: the former provides richer 3D information and reduces hand–object occlusions during grasping.

Table 9: **Success rates with different demonstrations.** Demonstrations are collected via teleoperation to grasp objects of different sizes (small vs. large) and from different directions (top vs. side). While directly replaying the demonstrations yields widely varying success rates across all objects, all policies learned by *DemoGrasp* achieve comparably high success rates.

| Demo. \ Success. | Demo Replay | RL Policy (Training Set) | RL Policy (Test Set) |
|---|---|---|---|
| small obj. + top | 75.29% | 96.24% | 82.74% |
| small obj. + side | 62.90% | 95.18% | 81.45% |
| big obj. + top | 7.23% | 95.02% | 82.46% |
| big obj. + side | 3.88% | 95.27% | 83.22% |

**Are 175 training objects enough?** Table 7 shows that when trained directly on the five test sets, the policy achieves an average performance gain of 2.4% relative to training on 175 objects and testing on these same test sets. This marginal gain suggests that universal grasping can be achieved with a small training set using *DemoGrasp*.

**Demonstration quality.** We study the effect of demonstration quality for *DemoGrasp*, using demonstrations that grasp different objects and approach from different directions. Results in Table 9 show that *DemoGrasp* consistently achieves high performance given any successful demonstration.

## 4 CONCLUSION

*DemoGrasp* is an RL framework for universal dexterous grasping that leverages a single demonstration to mitigate the exploration challenge. By formulating a single-step MDP with a compact action space for demonstration editing, *DemoGrasp* eliminates the need for complex reward shaping and can robustly achieve excellent success rates when trained on diverse robotic hand embodiments and object datasets. Extensive simulation experiments demonstrate that *DemoGrasp* achieves state-of-the-art success rates on multiple dexterous hands and object datasets. Real-world experiments show that *DemoGrasp* achieves robust zero-shot sim-to-real transfer via simple vision-based imitation learning, successfully grasping diverse real-world objects, including small and flat items that have remained challenging in tabletop settings for prior work. Taken together, the contributions of *DemoGrasp* not only establish a novel approach to dexterous grasping but also provide an easy-to-implement, robust RL framework for robotics research and applications.

ACKNOWLEDGMENTS

This work was supported by NSFC in part under Grant 62450001 and 62476008. The authors would like to thank the anonymous reviewers for their valuable comments and advice.

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

## A  LIMITATIONS AND FUTURE WORK

Although our method handles universal dexterous grasping in both simulation and the real world and extends to cluttered and instruction-based grasping, some advanced tasks still require nontrivial design—such as functional grasping and tightly cluttered scenes that require pre-grasp manipulation. In addition, while our policies exhibit some closed-loop regrasp capabilities through vision-based distillation, the policy learned in the RL stage is open-loop and cannot handle dynamic scenes or fine-grained manipulation. Future work could explore a trade-off between *DemoGrasp* and direct RL in the low-level action space to enable both efficiency and closed-loop behaviors. For example, by breaking demonstration trajectories into short segments and having the RL policy operate at the segment level.

Using pure visual observations may limit the performance of our method, particularly when the dexterous hand occludes the object in camera images, when the real-world object requires grasping with appropriate strength, or when the task demands precise grasping for downstream manipulation (e.g., grasping an electric screwdriver with the index finger pressing the button). Incorporating tactile feedback presents a promising future direction to mitigate these issues. Future work could include tactile data in the observations and design reward functions to achieve more precise grasping. Given the sim-to-real challenge posed by tactile sensors, another promising direction is to collect real-robot data with tactile sensing using our vision-based policy and then finetune it with the collected data to incorporate tactile information.

## B  RELATED WORK

**Universal Dexterous Grasping** is a fundamental task for robotic manipulation. Research approaches can be broadly classified into static grasp generation and dynamic grasping policy learning.

The goal of **grasp generation** is to synthesize hand and wrist poses of robust grasps, serving either as targets for grasping motion planning (Wang et al., 2023; Zhang et al., 2024a) or as auxiliary information for policy learning (Xu et al., 2023). Recent studies have explored various strategies: Weng et al. (2024) employ diffusion models to generate grasping poses; Zhong et al. (2025b) enhance diffusion-based generation by incorporating physical constraints during sampling; Wei et al. (2024) train a grasp generation model for multiple hand embodiments; Jian et al. (2025) leverage contact and affordance priors retrieved from existing grasp examples; and Chen et al. (2025a) propose a model that generates grasp poses conditioned on predefined grasp taxonomies. Beyond synthesizing physically plausible grasps, recent research in language-guided grasping (He et al., 2025; Li et al., 2024) introduces the additional challenge of learning a joint distribution between natural language and dexterous grasp poses. However, deploying grasp-generation models still requires manual motion-planning design and faces challenges in tabletop settings when a collision-free grasp trajectory does not exist.

**Policy-learning** methods aim to learn closed-loop grasping policies over low-level robot actions, enabling direct deployment on real robots and avoiding explicit collision handling and path planning. Yet training universal grasping policies via RL is challenging due to high-dimensional, long-horizon exploration, the need to optimize across diverse objects simultaneously, and the gap for closed-loop sim-to-real transfer. Prior work has explored curriculum learning (Xu et al., 2023; Chen et al., 2025b), policy distillation (Wan et al., 2023; Wang et al., 2025; Yuan et al., 2025a), residual learning (Huang et al., 2025; Zhao et al., 2025), and object-geometry representation (Zhang et al., 2025a;b) to mitigate the RL exploration burden in simulation.

Recent advances include UniGraspTransformer (Wang et al., 2025), which proposes a Transformer-based distillation method that can distill thousands of single-object policies without significant performance loss, achieving SOTA performance on DexGraspNet. However, it still struggles with sim-to-real transfer. RobustDexGrasp (Zhang et al., 2025b) trains a performant policy for universal dexterous grasping using geometry representation and contact prediction, achieving sim-to-real transfer on a large number of objects. However, the method requires high-quality point cloud observations and is evaluated only on the Allegro Hand, limiting its ability to grasp small and thin objects. ClutterDexGrasp (Chen et al., 2025b) explores grasping in cluttered scenes using a similar geometry representation with a contrastive distance reward function, but it is evaluated on a specific hand and cannot grasp small objects in real-world environments. DexGraspVLA (Zhong et al.,

2025a) trains an RGB-based grasping policy solely from real-world teleoperation data but is limited to large objects with a single grasp pose, making it costly to acquire training data and limiting object generalization. DextrAH-RGB(Singh et al., 2024) and DextrAH-G (Lum et al., 2024) explore sim-to-real techniques using RGB or depth images, but their real-world evaluations are limited to approximately ten regular-sized objects.

In our work, we address the challenge of designing high-performance, efficient RL pipelines for grasping policy learning. We mitigate the exploration burden by introducing a single demonstration and reformulating the problem as a single-step MDP. As a result, our method **extends to any dexterous hand embodiment** without hyperparameter tuning and achieves **performant sim-to-real transfer** across a wide variety of unseen objects, including **small and thin ones**, using only simple **RGB observations**.

**Learning Robotic Manipulation from Demonstrations** is an active research direction that leverages human priors from demonstration data to facilitate robot learning. Imitation learning methods directly learn robot policies by imitating human teleoperated data (Zhao et al., 2023; Fu et al., 2024; Chi et al., 2023) or human manipulation data (Luo et al., 2025; Bi et al., 2025), but require a large number of high-quality trajectories. Some works (Mandlekar et al., 2023; Jiang et al., 2025; Xue et al., 2025) improve data efficiency for imitation learning by synthesizing multiple demonstration trajectories from a single demonstration. Demonstrations can also be used to augment RL, serving as reward signals (Chen et al., 2024; Zhou et al., 2024; Li et al., 2025), auxiliary losses (Qin et al., 2022), and training data in the replay buffer (Luo et al., 2024). In our work, we introduce a single successful grasp demonstration to facilitate RL for universal grasping. By appropriately transforming wrist and hand poses, this single demonstration is augmented into diverse grasp trajectories for arbitrary objects, enabling a universal policy.

## C OBJECTS USED IN EXPERIMENTS

Figure 5 shows the 110 objects used in our real-world experiments. Figure 6 shows samples from each dataset used in our simulation experiments.

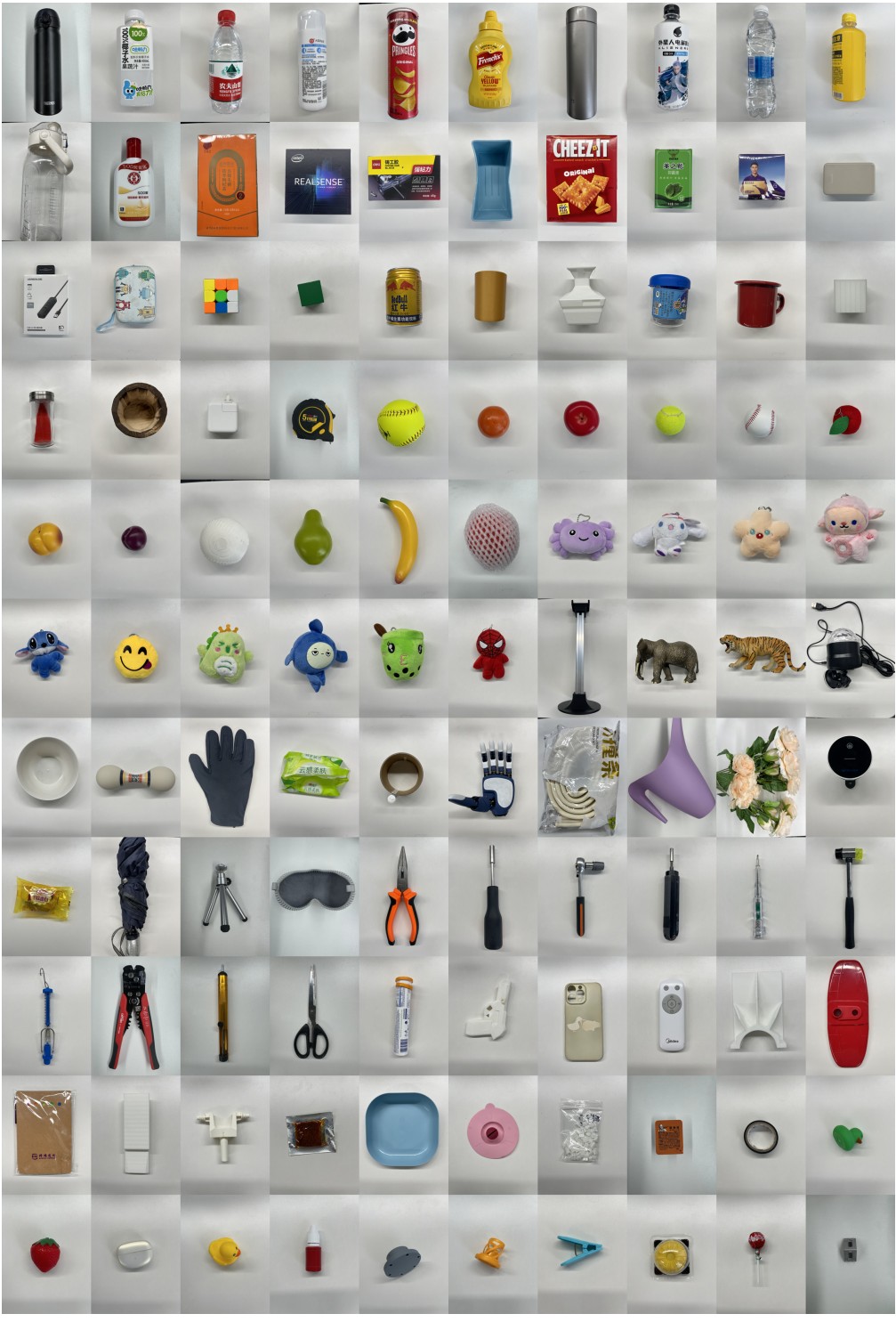

Figure 5: Objects used in real-world experiments.

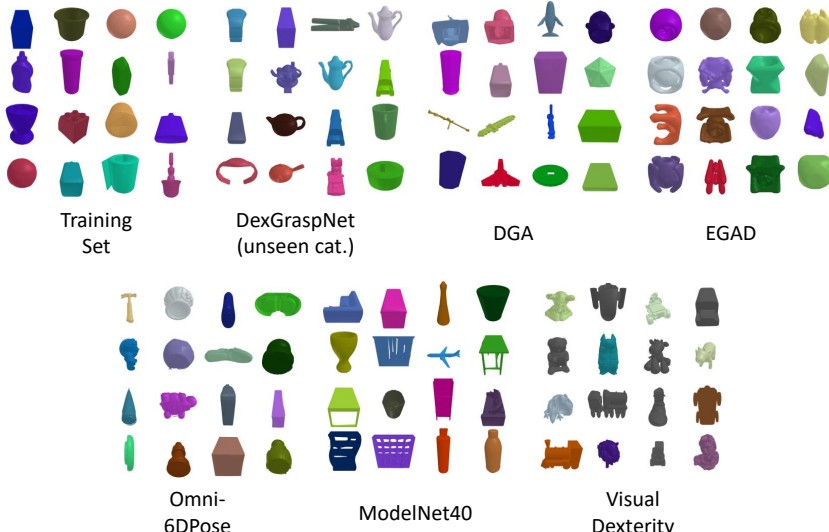

Figure 6: Snapshots of the training and test datasets used in our experiments. 16 objects are randomly sampled from each dataset for visualization.

# D ADDITIONAL RESULTS

## D.1 QUANTITATIVE RESULTS

Table 10: Success rates of DemoGrasp for various robotic embodiments across all object datasets.

| Embodiment | Training Set | DexGraspNet (Unseen Cat.) | DGA | EGAD | Omni-6DPose | ModelNet40 | Visual Dexterity |
|---|---|---|---|---|---|---|---|
| FR3 + Gripper | 90.21 | 81.10 | 30.71 | 49.38 | 66.56 | 79.46 | 83.49 |
| FR3 + DClaw | 96.96 | 94.17 | 79.28 | 96.37 | 74.63 | 61.16 | 97.94 |
| UR5 + Allegro | 92.93 | 94.18 | 74.40 | 96.75 | 82.24 | 75.58 | 97.80 |
| FR3 + Inspire | 96.24 | 97.5 | 65.62 | 97.88 | 85.04 | 80.13 | 99.13 |
| UR5 + Schunk | 95.46 | 87.29 | 47.97 | 90.70 | 78.19 | 73.43 | 88.59 |
| Shadow | 97.43 | 89.75 | 60.99 | 94.58 | 74.84 | 71.88 | 93.67 |
| FR3 + Shadow | 95.33 | 87.40 | 59.90 | 93.64 | 76.51 | 67.48 | 93.40 |

## D.2 QUALITATIVE RESULTS

Figure 7 shows real-world grasp trajectories for arbitrary objects. Figure 8 shows real-world tests under complex, randomized scene configurations using a language-conditioned policy.

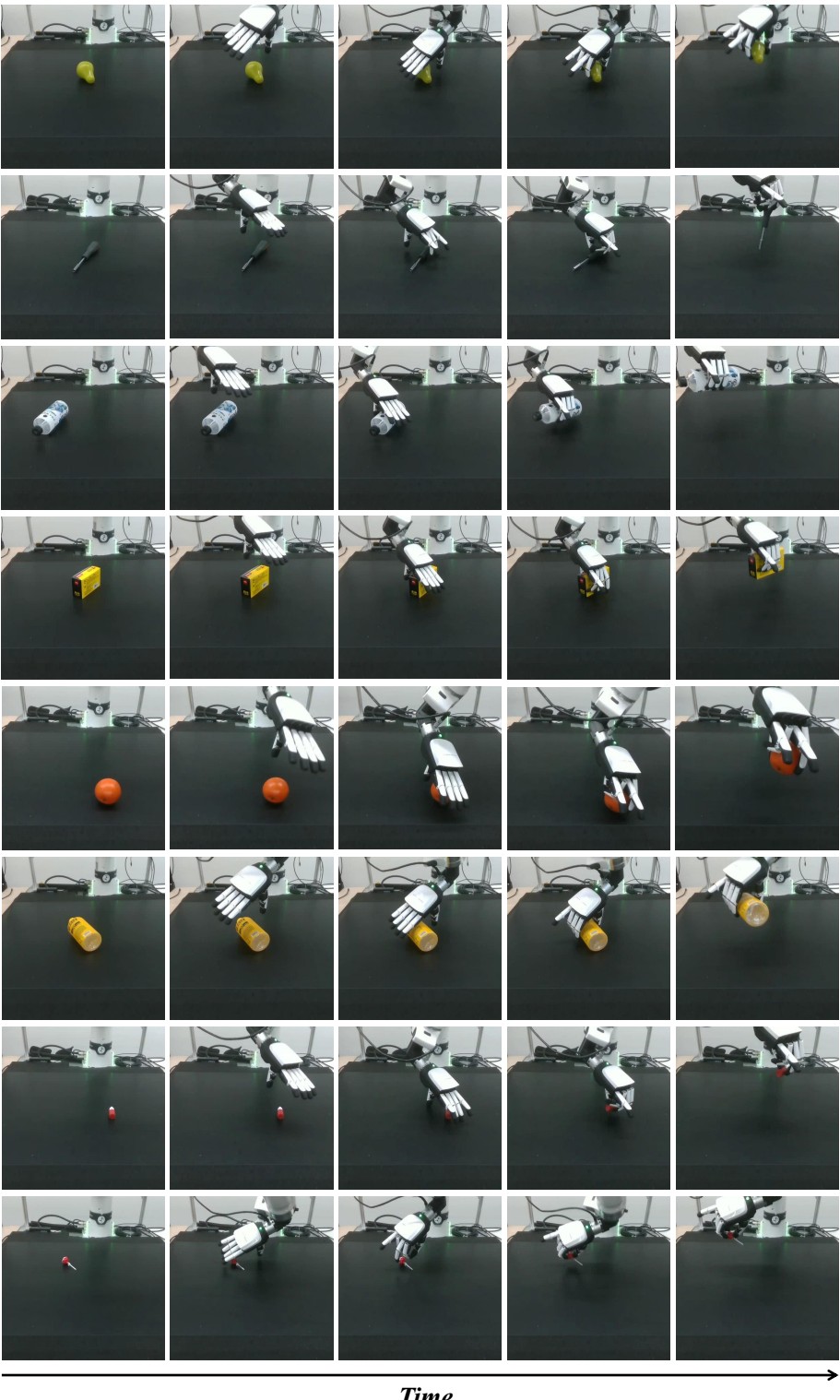

*Time*

Figure 7: Grasping trajectories from real-world tests. *DemoGrasp* learns distinct finger poses for large, small, and thin objects to maximize expected success. Slight robot–table contact is leveraged to grasp tiny objects (second-to-last row). A regrasp behavior emerges when an attempt fails (last row).

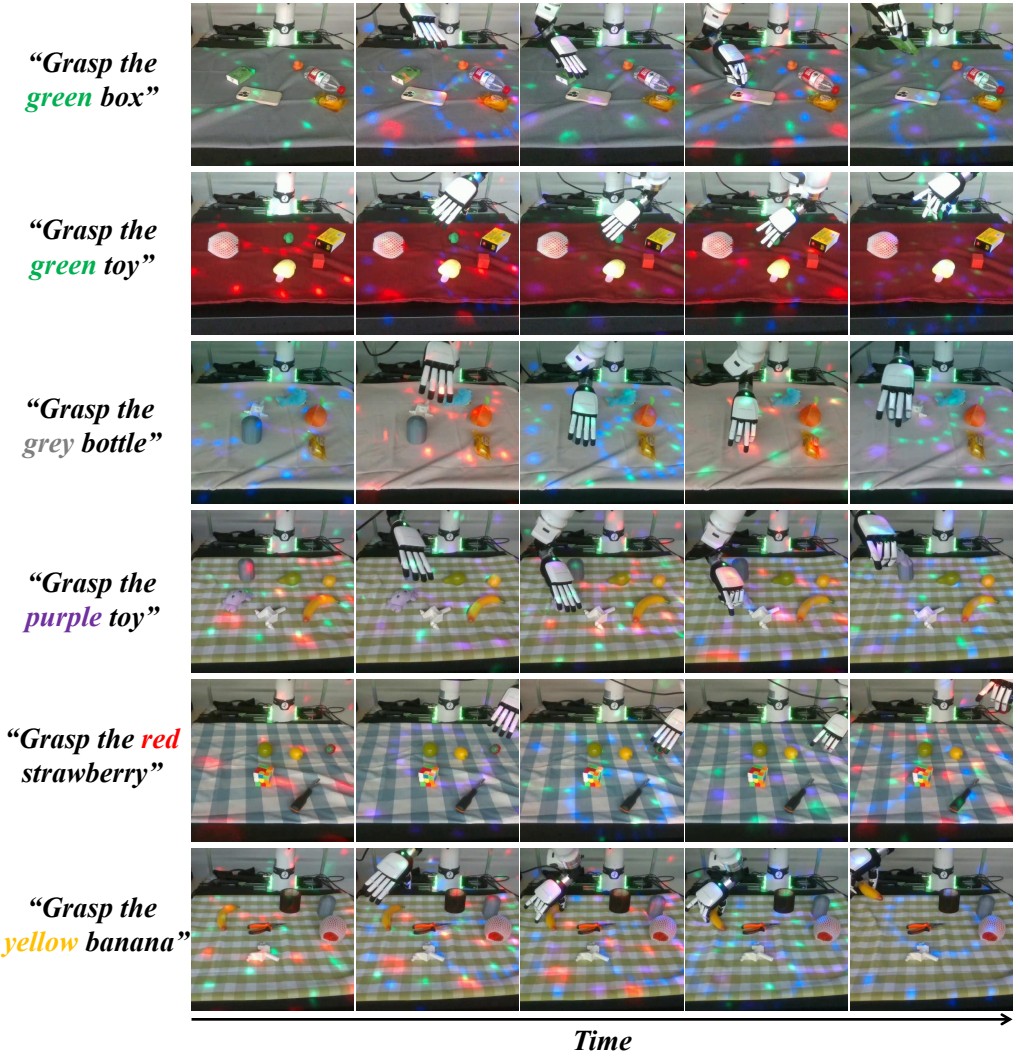

Figure 8: Real-world tests of the language-conditioned *DemoGrasp* policy in cluttered scenes with randomized object positions, language instructions, backgrounds, and lighting conditions.

### D.3 THE NECESSITY OF TWO-STAGE TRAINING

There are several reasons motivating us to adopt a two-stage training pipeline (state-based RL for data collection followed by vision-based imitation learning) rather than directly training a vision-based RL policy.

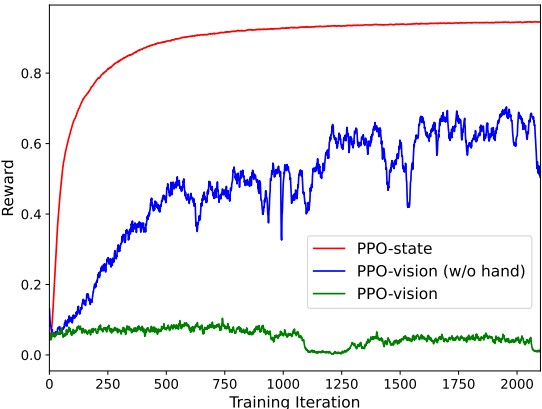

Figure 9: Comparison of RL training curves between the state-based policy (7000 parallel environments) and the vision-based policy (175 parallel environments). "w/o hand" denotes a reduced action space that includes only end-effector transformations, while others use the full action space.

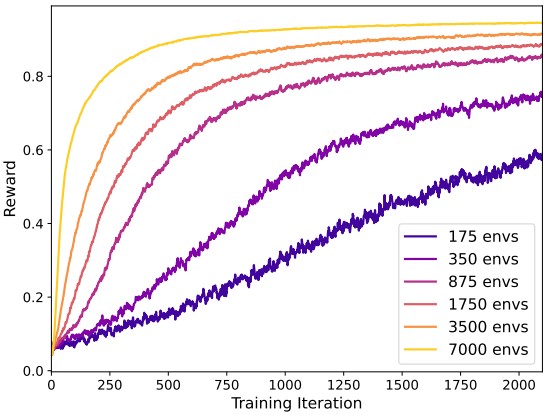

Figure 10: Comparison of RL training curves with different numbers of parallel simulation environments, ranging from 175 to 7000.

First, large-scale parallel simulation is necessary for efficient RL training on robotic tasks. If we directly train a vision-based RL policy, rendering RGB images consumes substantial GPU memory, resulting in fewer than 200 parallel environments on a single RTX 4090 GPU. In contrast, state-based training without rendering supports up to 7000 parallel environments. The large number of parameters introduced by the vision encoder also increases training cost. Therefore, given the same computation budget, state-based RL allows significantly more parallel environments and leads to better training performance. To verify this, we implement a vision-based RL pipeline in which the policy takes two RGB images processed by a pretrained ResNet-18 encoder. Figure 9 shows that state-based RL, which uses 40 times more parallel environments than vision-based RL, outperforms it in both sample efficiency and final performance. Figure 10 compares training curves under different numbers of

parallel environments and confirms the necessity of large-scale simulation (typically more than 1000 environments) for RL.

Second, our method reformulates RL as demonstration editing, where actions modify wrist transformations and hand delta angles. However, real-world deployment requires closed-loop control in the robot action space (per-timestep end-effector and hand joint targets). Therefore, a two-stage pipeline is necessary: RL optimizes in the demonstration-editing action space, and imitation learning distills the behavior into a policy that directly outputs robot actions. Online distillation methods such as DAgger are not applicable in this setting because they require the teacher policy and the student policy to share the same action space. For this reason, we collect successful rollouts from the teacher policy and train the student policy using imitation learning on the offline dataset. This two-stage RL+BC approach is also used in prior work (Wang et al., 2025; Yin et al., 2025; Liu et al., 2025).

### D.4 TRAINING CURVES

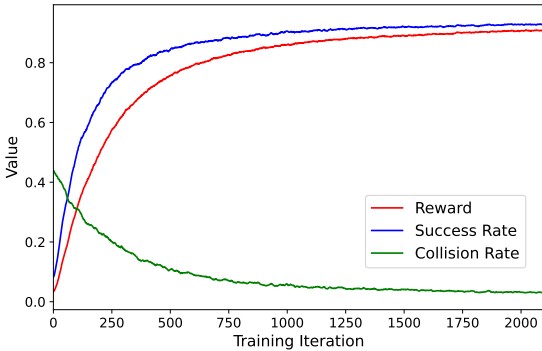

Figure 11: RL training curves for the Inspire hand, showing total reward, success rate, and collision rate.

## E IMPLEMENTATION DETAILS

### E.1 DATASETS AND SIMULATION

In Sec. 3.2, we compare our method against prior approaches for universal dexterous grasping, all trained and evaluated on DexGraspNet (Wang et al., 2023). Following their protocol, we adopt the same split consisting of a training set and two test sets. The training set contains 3,200 objects spanning diverse shapes and categories. The first test set includes 141 objects from categories seen during training but with novel shapes not present in the training set. The second test set contains 100 objects whose shapes and categories are both unseen during training, thereby assessing generalization to entirely novel object classes.

For the experiments in Sec. 3.3 and the real-world experiments, we adopt a smaller training set. Specifically, we randomly sample 100 objects from the DexGraspNet training set and add 75 objects from YCB to increase shape and category diversity, yielding 175 training objects in total. We evaluate the learned policies on multiple datasets, including DexGraspNet (the 100 unseen-category objects) (Wang et al., 2023), DGA (Zhong et al., 2025b), EGAD (Morrison et al., 2020), Omni6DPose (Zhang et al., 2024b), ModelNet40 (Wu et al., 2015), and Visual Dexterity (Chen et al., 2023). The results demonstrate robustness across diverse embodiments and datasets.

We use IsaacGym (Makoviychuk et al., 2021) as our simulation platform. For state-based policies, we sample 512 points from mesh vertices and surfaces for each object to form its full point-cloud representation. We increase the resolution of VHACD decomposition to 300K so that the objects' collision meshes in the simulator closely match the original meshes. We use a hierarchical position controller for arm and hand control, where a high-level controller receives target joint positions and interpolates from the previous target to the new target, sending a smooth trajectory to a low-level PD

controller. We set the PD controller stiffness to large values and limit the step between consecutive target joint positions to produce smooth motion and accurately track the policy's actions. For robot hands mounted on arms, we use inverse kinematics to convert the policy's end-effector action outputs into target joint positions for the controller. Detailed simulation and control parameters are provided in Table 11.

Table 11: Simulation parameters.

| Name | Value |
|------|-------|
| Low-level control frequency | 60 Hz |
| Policy control frequency | 3 Hz |
| Simulation substeps | 2 |
| Object friction coefficient | 1.0 |
| Maximum arm angular velocity | 1.57 rad/s |
| Maximum hand angular velocity | 6.28 rad/s |
| Maximum hand joint effort | 1.0 |
| Arm joint stiffness $K_p^{\text{arm}}$ | 16000 |
| Arm joint damping $K_d^{\text{arm}}$ | 600 |
| Hand joint stiffness $K_p^{\text{hand}}$ | 600 |
| Hand joint damping $K_d^{\text{hand}}$ | 20 |

At the beginning of each replay of the edited demonstration, we require a motion planner that can smoothly and accurately move the robot from its initial pose to the starting pose in the edited demonstration within the initial object frame. To achieve this, we use an interpolation-based motion planner for the robot end-effector, where positions are linearly interpolated with a maximum step size of 0.04 m, and rotations are interpolated using SLERP with a maximum step size of 0.1 rad. Each interpolated pose serves as a target for the PD controller, which is executed for 20 simulation steps to reach the target. For the dexterous hand, we directly set the action to the initial joint positions from the demonstration. We verify that the robot can follow the action sequence with minimal tracking error using this motion planner in both simulation and the real world.

### E.2 TRAINING RL POLICIES

The demonstration-editing policy is trained jointly across all objects in the training set using Proximal Policy Optimization (PPO) (Schulman et al., 2017). Full object point clouds are encoded with PointNet (Qi et al., 2017), and the resulting 128-dimensional features are concatenated with the end-effector pose and the initial object pose in the world frame. This combined vector is then fed into the actor and critic networks, implemented as MLPs with hidden layers of sizes [1024, 1024, 512, 512] and ELU activations (Clevert et al., 2015). For the final action output layer, we apply $\tanh$ to bound outputs to $[-1, 1]$, then rescale to the allowed editing ranges: end-effector translation $\Delta\text{xyz} \in [-0.05, 0.05]$ m; end-effector Euler angles $\Delta\text{rpy} \in [-1.57, 1.57]$ rad; and delta hand joint angles $\Delta\mathbf{q} \in [-1, 1]$ rad. The hyperparameters used for training are summarized in Table 12. Training converges within 24 hours on a single NVIDIA RTX 4090 GPU.

### E.3 TRAINING VISION-BASED POLICIES

We collect 35,000 trajectories with the trained RL policy across all training objects and retain the successful trajectories to train the vision-based policy.

To bridge the sim-to-real gap in RGB images, we align camera and table configurations in simulation with those in the real world and apply broad domain randomization. We first estimate camera extrinsics via hand–eye calibration and set both camera intrinsics and extrinsics in simulation to match the real hardware. We then randomize camera extrinsics by adding uniform noise in the range $[-0.02, 0.02]$. We fetch 300 background images from RoboTwin (Mu et al., 2024) to sample table textures. For objects, we randomly sample colors and apply textures from 100 material images in FMD (Sharan et al., 2014). We use three point lights and randomize their intensity and ambient parameters within $[0.1, 0.8]$ to create varied lighting conditions. We also translate the table randomly within $[-0.05, 0.05]$ m to increase background variation. For ablation studies using depth images, we

Table 12: **Hyperparameters for RL.**

| Name | Value |
|---|---|
| Parallel environments | 7,000 |
| Initial actor Gaussian std. | 0.8 |
| Learning rate | 3e-4 |
| PPO clip range ($\epsilon$) | 0.2 |
| Gradient-norm clip | 1.0 |
| Observation clip range | 5.0 |
| Episode length for RL | 1 |
| Execution steps for demo replay per episode | 40 |
| Rollout steps per iteration | 1 |
| Update epochs per iteration | 5 |
| Minibatches per epoch | 4 |

follow HERMES (Yuan et al., 2025b) to augment the rendered depth observations. We randomize the depth range, add Gaussian blur and Gaussian noise, randomly set 1% of depth pixels to zero, and blend each frame with a random depth image from the NYU-Depth-v2 dataset (Nathan Silberman & Fergus, 2012) using $\alpha = 0.005$.

We adopt the GR00T-N1.5 (Bjorck et al., 2025) architecture, consisting of a pretrained Vision Transformer (ViT) encoder and a flow-matching action head. We do not use the pretrained weights from GR00T-N1.5; instead, we train the action head from scratch and fine-tune the pretrained ViT. The model is trained for 100k iterations on four NVIDIA A800 GPUs, taking 16 hours.

To train a language-conditioned policy for grasping in cluttered scenes, we sample distractor objects and construct language instructions during data collection. For each environment, we load 10 objects sampled from the training dataset. For each episode, we uniformly sample $n \in [0, 10]$ objects among them as distractors and initialize them at random poses on the table. We use the instruction template "Grasp the {COLOR} {OBJECT_NAME}." where {COLOR} is the color name sampled for the task object and {OBJECT_NAME} is its instance name from the dataset. We use the pretrained vision-language model open-sourced by GR00T-N1.5 to process language instructions during imitation learning, keeping its weights frozen. In real-world tests, we use the true color and name of the object (which may be unseen during training) to prompt the model.

### E.4 REAL-WORLD EXPERIMENTS

We conduct real-world experiments using a Franka Research 3 robot arm with an Inspire Hand. The world frame is defined at the arm's base frame. Two RealSense D435i cameras are placed at fixed viewpoints to capture RGB or depth images for the vision-based policies. Figure 12 illustrates the hardware setup and camera views.

## F  AN ILLUSTRATION OF DEMONSTRATION EDITING

Figure 13 illustrates how the learned policy edits the demonstration of grasping a little duck to grasp a plate.

## G  USE OF LARGE LANGUAGE MODELS

In this work, the large language model (LLM) is used exclusively for text polishing. Its role is limited to refining the linguistic quality of the textual content, with no involvement in the method or experimental results.

## H  OPEN-SOURCE COMMITMENT

We are committed to open-sourcing the code upon paper publication.

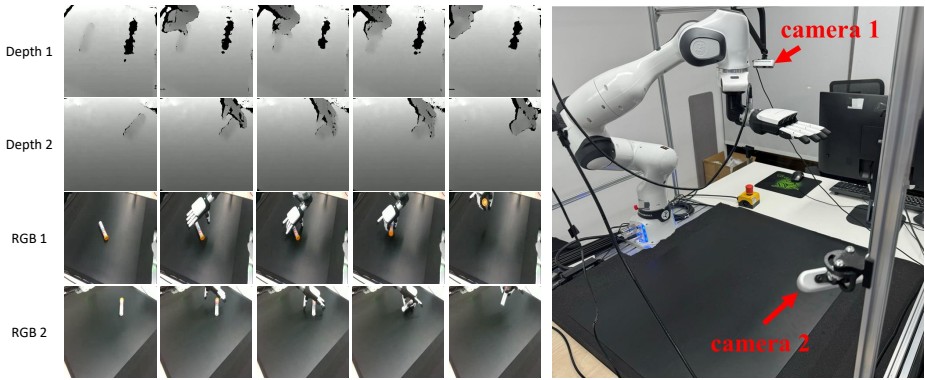

Figure 12: Example images from different camera sensors used in our experiments (left) and our hardware setup (right).

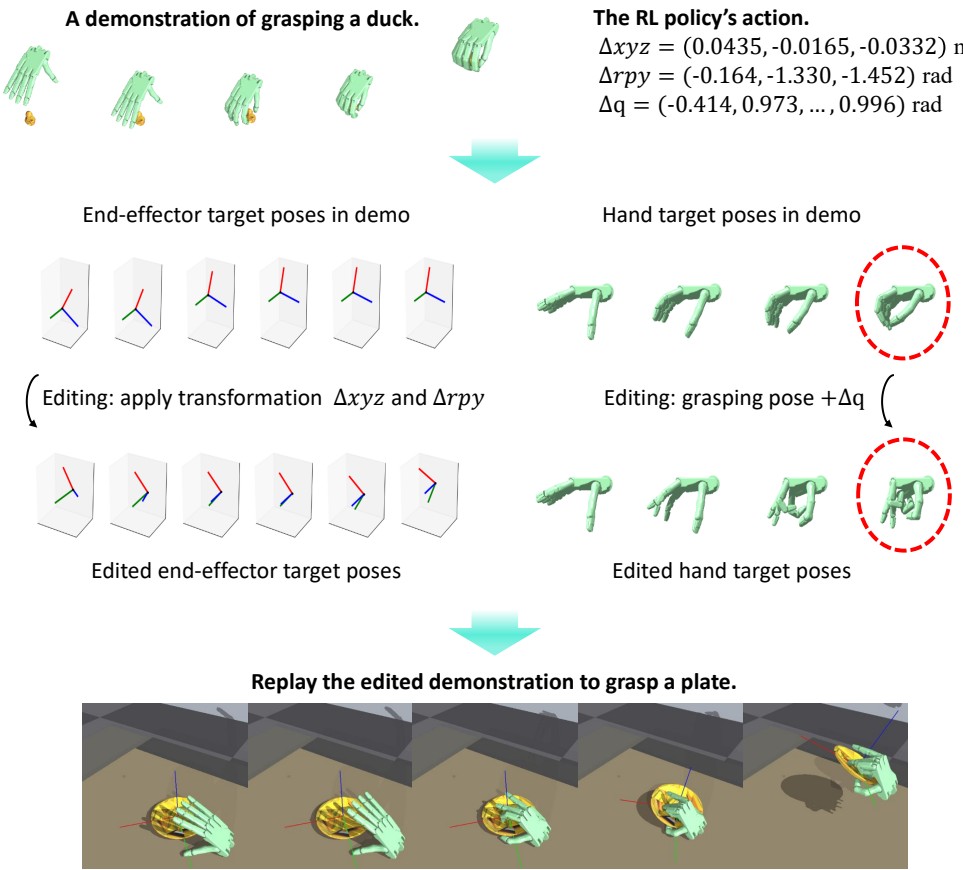

Figure 13: An illustration of demonstration editing using the learned RL policy.

