# OpenReview forum: "DemoGrasp: Universal Dexterous Grasping from a Single Demonstration"
_ICLR.cc/2026/Conference — ICLR 2026 Poster_

### Official Review · Reviewer_kYw5 · 2025-11-01

**Soundness:** 3
**Presentation:** 3
**Contribution:** 2
**Rating:** 6
**Confidence:** 5

**Summary:**

This paper presents a method for training dexterous grasping policies from a single demonstration in simulation. The key idea is to frame the problem as demonstration editing, where a reinforcement learning (RL) policy learns to generate residual corrections that enable grasping of novel objects. This formulation helps address the exploration challenge and leads to more efficient learning. Extensive real world evaluations are carried out to validate the method.

**Strengths:**

The paper is well-written and easy to follow.

The proposed method is conceptually sound and addresses an important problem in dexterous manipulation.

**Weaknesses:**

1. It is unclear what range of novel objects this approach can effectively handle given only a single demonstration on one object. For example, if the demonstration involves grasping a bottle, it may be difficult to transfer that experience to grasping a flat plate. It would be valuable to include experiments illustrating which types of object transfers are successful, and to discuss how such transferability might be predicted a priori.

2. The computational efficiency of the proposed method appears to remain high compared to sampling-based or template-based approaches. Including quantitative comparisons of computational cost and a discussion of when this approach would be preferable to these alternatives would strengthen the paper.

3. The high-level idea is not that surprising, given recent papers like DexMachina.

**Questions:**

See Weakness 1. I would like to see more discussion on this.

---

> ### Author Response · Authors · 2025-11-23
> **Thanks for your review! Here, we respond to your comments and address the issues. We hope to hear back from you if you have further questions!**
>
> **Q1.** What range of novel objects can the method handle given one demonstration?
>
> **A1.** As shown in our experiments, the trained policy handles all objects with high success rates (across seven object datasets in simulation and 110 real-world objects, without cherry-picking), using only one demonstration for training. The reason for such strong transferability is that our demonstration-editing action space allows for significant variation in grasping style, making it **adaptable to any object**. Specifically, we use end-effector transformations to adjust the grasping offset and orientation within the object frame, and hand delta actions to modify the hand's grasping pose.
>
> We have added an illustration in Appendix F (Figure 13) to visualize how our RL policy edits a demonstration of grasping a small duck to grasp a plate. The figure shows that the policy adjusts the x-axis offset significantly to make the hand grasp the edge of the plate and corrects the hand’s pose to achieve a tighter grasp.
>
> Since the RL action space is defined by the range of wrist translation offsets, Euler angles, and hand delta poses, **the capability of transferring to novel objects is predictable**. For example, in our experiments, we set the translation range to [-5, 5] cm and the Euler angle range to $[-\pi/2, \pi/2]$. This allows the policy to explore possible grasps within this range around the object’s geometric center, enabling it to handle objects with grasping diameters of up to 10 cm.
>
>
> **Q2.** Computational cost compared with sampling-based approaches.
>
> **A2.** We would like to clarify that the efficiency of our RL-based method actually outperforms sampling-based grasp generation methods when generating a large dataset of tabletop grasping trajectories. Our RL method can optimize for all objects simultaneously in parallel simulation, while sampling-based methods take time to optimize for each object and initial pose individually.
>
> We compare the efficiency of our method with BODex [1], the SOTA sampling-based method for dexterous grasp generation. We evaluate generating successful grasping trajectories for DexGraspNet objects using the Shadow Hand on a single RTX 4090 GPU. Our method takes less than 24 hours to train an RL policy for 3,200 objects in parallel. Collecting a successful trajectory for any object using the learned policy incurs very low cost (it can collect 7,000 trajectories in IsaacGym within 30 seconds). In contrast, BODex takes 3 seconds to optimize the grasping pose and generate a valid trajectory for each object at a specific initial pose.
>
> The table below compares the time cost to generate a tabletop grasping dataset with different numbers of trajectories:
>
> | Num. Trajectories | 1K |  10K | 30K | 100K | 1M |
> |----------|---------|----------|---------|---------|---------|
> | BODex | 1h | 8h | 24h | 80h | 800h |
> | DemoGrasp | 24h | 24h | 24h | 24h | 25h |
>
> When more than 30,000 trajectories are required, our RL-based method outperforms the sampling-based method in computational efficiency.
>
> [1] Chen et al., BODex: Scalable and Efficient Robotic Dexterous Grasp Synthesis Using Bilevel Optimization, ICRA 2025.
>
>
> **Q3.** About the novelty of the high-level idea compared with papers like DexMachina.
>
> **A3.** While papers like DexMachina also learn RL policies to correct actions from demonstrations, our work introduces a novel methodology for learning these corrections, leading to stronger transferability to objects different from those in the demonstration.
>
> Previous work typically learns a **residual policy in the robot action space**, where the RL policy’s action is numerically added to the base action to control the robot. This approach faces a trade-off [2] in selecting the range of residual actions: if the range is too large, RL fails to leverage the base actions from the demonstration; if the range is too small, RL cannot adapt to objects with significantly different shapes.
>
> In contrast, our study formulates RL as an **object-centric demonstration editing** problem. We introduce an action space that includes SE(3) transformations to modify the end-effector’s trajectory and hand delta actions to adjust the hand’s grasp pose, allowing for **greater flexibility** in adapting to different objects. By reducing the horizon to one step and replaying the edited demonstration in an object-centric manner, our method achieves high sample efficiency, even with a wide action space.
>
> [2] Yuan et al., Policy Decorator: Model-Agnostic Online Refinement for Large Policy Model, ICLR 2025

---

### Official Review · Reviewer_mozF · 2025-11-01

**Soundness:** 2
**Presentation:** 3
**Contribution:** 2
**Rating:** 6
**Confidence:** 4

**Summary:**

This paper presents a method to learn dextrous grasping from just a single demonstration. The key differentiator with existing approaches is to formulate the problem as a 1 step markov decision process where the policy predicts a residual trajectory acting as a demonstration editor. The entire trajectory is treated as a single action executed using a lower level motion planner. This makes exploration and reward design much more simpler than imitating the entire trajectory. Once a state-based policy is trained in simulation, a vision-action dataset is collected to then learn a vision-based policy using behavior cloning.
The results show generalization to novel objects as well as zero shot transfer to real world robots.

**Strengths:**

The paper is well structured and easy to follow. It provides sufficient background, comparison to existing literature and clearly establishes the motivation for the paper.

The paper provides sufficient details about training, provides comprehensive quantitative comparison against relevant baselines across different object datasets and robot embodiments. The results and the supplementary videos show effective grasp success rates.
The paper also presents in depth ablation studies that sheds light on the necessity for RL instead of sampling based methods, the action space used in RL, camera configurations as well as demonstration quality. The proposed method performs better than baselines in all scenarios.

An additional strength of the paper is real-world results which is the ultimate test for any robot learning method.

**Weaknesses:**

While the results are promising, I think the biggest weakness is the lack of generalizability of the proposed method to more dynamic environments where a reactive policy might be required to avoid collisions or adjust a grasp mid motion.

Adding details about how the language conditioned policy is learned can be extremely beneficial for a reader. Similarly, more details on the lower level motion planner that executes the trajectory would be really helpful.

**Questions:**

1) The problem formulation is not too clear - In section 3.1, the authors mention that the goal of the policy is to maximize the discounted sum of returns (like standard RL), and the equation suggests that the horizon “T” is the length of the demonstration trajectory. But this is at odds with the single step MDP that the authors claim in the beginning. Even in table 12, the episode length is set to 1. How exactly is the problem formulated and what is the discount factor?

2) What is the motion planner used? How robust is the policy to the motion planner?

3) The vision based policy is closed loop per grasp attempt, correct? What is the frequency of that loop?

4) It would also be great to know how tactile feedback can be incorporated into this framework in future work section.

---

> ### Author Response · Authors · 2025-11-23
> **Thanks for your review! Here, we respond to your comments and address the issues. We hope to hear back from you if you have further questions!**
>
> **Q1.** Implementation details for the language-conditioned policy and the motion planner. How robust is the policy to the motion planner?
>
> **A1.** We thank the reviewer for the valuable suggestions. We have added details on the implementation of the language-conditioned policy in Appendix E.3 and the implementation of the motion planner in Appendix E.1.
>
> To smoothly and accurately move the robot from its initial pose to the starting pose in the edited demonstration, we use an interpolation-based motion planner for the robot’s end-effector. Positions are linearly interpolated with a maximum step size of 0.04 m, and rotations are interpolated using SLERP with a maximum step size of 0.1 rad. Each interpolated pose serves as a target for the PD controller, which is executed for 20 simulation steps to reach the target. For the dexterous hand, we directly set the action to the initial joint positions from the demonstration. We verify that the robot can follow the action sequence with minimal tracking error using this motion planner in both simulation and the real world.
>
> To test the robustness of the learned policy, we vary the interpolation step size of the motion planner and the stiffness of the PD controller in simulation. The table below shows the policy's success rate on training objects with different planners and controller parameters:
>
> | stiffness \ interpolation step size |	0.5x | 1x | 2x |
> |----------|---------|----------|---------|
> | 0.5x | 0.959 | 0.958 | 0.955 |
> | 1x | 0.961 | 0.951 | 0.958 |
> | 2x | 0.958 | 0.952 | 0.951 |
>
> The results show that variations in motion planner and controller parameters do not significantly affect policy performance, indicating that our setup ensures the robot can closely follow the target poses in actions.
>
> **Q2.** The confusion about the MDP horizon in the problem formulation.
>
> **A2.** We apologize for the confusion caused by the change in the MDP formulation. In Section 2.1, we provide a common problem formulation for dexterous grasping, which is a T-step MDP. Section 2.2 introduces demonstration editing to reduce the complexity of the problem. Finally, **in Section 2.3, the problem is reformulated as a single-step MDP.** In our final method, for each episode, the RL policy only takes one action (demo editing parameters). The edited demonstration is replayed for T steps to interact with the environment, and a single reward is given after replay. As a result, the discount factor is not needed, and RL optimizes for the single-step reward.
>
> To reduce confusion, we have revised Section 2.1 better to guide the reader toward the final single-step MDP formulation. In Table 12, we distinguish between the episode length for RL (which is 1) and the execution steps for demo replay within each episode (which is T=40).
>
> **Q3.** What is the frequency of the vision-based policy?
>
> **A3.** The closed-loop vision-based policy runs at 3 Hz. It takes images and the robot pose (end-effector pose and hand joint angles) as input and outputs a target robot pose. This target pose is sent to the low-level PD controller, which runs at 60 Hz.
>
> **Q4.** Incorporating tactile feedback in future work.
>
> **A4.** We have added a paragraph in Appendix A discussing the limitations of a purely vision-based policy and potential directions for incorporating tactile sensors in future work.

---

### Official Review · Reviewer_QqKj · 2025-11-01

**Soundness:** 4
**Presentation:** 2
**Contribution:** 3
**Rating:** 8
**Confidence:** 4

**Summary:**

The paper shows framework for dexterous grasping policies using RL with single demonstration augmented. The policy could grasp a huge amount of objects and its appreciate the author conduct steps for sim-to-real, instead of simply reply the trajectory in simulation. The trajectory dataset and the sim-to-real for these trajectories could be a good contribution.

**Strengths:**

Appreciate the sparse reward and simple collision penalty. Easy to read and follow the idea. The presentation is good. The experiments are comprehensive.

**Weaknesses:**

Minor weakness:

1. the abstract contain many contents but not organize well. It get reader confuse that the author using RL in simulation but directly switch to imitation learning in the real-world without any context.

2. line 090, please specify what is the "prior methods" mentioned, eg cite the corresponding paper.

3. line 105-106, please specify what is "previous state-of-the-art-methods" here.

4. Front size in figure 1 is too small to recognize.


Major weakness:

1. when perform sim-to-real transfer, the author select the successfully RL rollout, then apply an additional training stage and apply domain randomization in imitation stage. why not directly train a vision based policy and directly transfer to the real.

2. For two stage training, why select stage based RL and do a selection and then do BC? Have you ever research train a state based RL first and then perform a teacher-student pipeline to distill a vision policy, which can also be able to transfer to real.

3. For the sim to real, the author mention the domain randomization for the RGB and depth. Have you apply any method on robot control side. eg. simulation hand is usually more compliant than real hand, especially for the inspire hand, which may like to broken if the collision is too large. Also for the arm, have you notice any control gap between sim and real.

4. Would be more interested the pipeline and application to extend to beyond grasping task.

5. How much randomization for the object initial pose? For a simple demo augumentation, does the larger randomization area decrease the max success rate of RL?

6. While appreciate conduct the BC for sim-to-real, more experiments to justify the BC training would make experiments more comprehensive. Eg. better generalization, or better design choice, or help on implementation.

**Questions:**

See weakness above

---

> ### Author Response · Authors · 2025-11-23
> **Thanks for your review! Here, we respond to your comments and address the issues. We hope to hear back from you if you have further questions!**
>
> **Q1.** Paper writing issues.
>
> **A1.** We thank the reviewer for the valuable suggestions and have revised the mentioned points in the paper accordingly:
> - We have revised the Abstract to clarify the role of vision-based imitation learning in our pipeline.
> - Line 90 and 106: We have cited the mentioned prior methods and the SOTA method on DexGraspNet.
> - We have adjusted the font sizes in the legend of Figure 1.
>
> **Q2.** About the options of (1) directly training vision-based RL; (2) applying teacher-student online distillation. Experiments to justify the use of BC.
>
> **A2.** We have added a section in Appendix D.3 that explains why we adopt an RL + BC pipeline (rather than directly using vision-based RL or using RL + online distillation) and provides experimental analysis. Please refer to the revised paper for further details.
>
> Here is a brief summary:
> - Large-scale parallel simulation is essential for efficient RL training in robotic tasks. Rendering RGB images in **vision-based RL consumes substantial GPU memory**, limiting the number of parallel environments that can be run on a single RTX 4090 GPU to around 200. In contrast, state-based RL can run 40 times more parallel environments, leading to better performance. Figure 9 demonstrates that state-based RL significantly outperforms vision-based RL with the same computational budget, and Figure 10 shows the importance of having more than 1000 parallel environments for effective RL training.
> - **Online teacher-student distillation is not applicable in our framework** because it requires the teacher and student policies to share the same action space. Our framework reformulates RL as demonstration editing, meaning the RL action space is different from the robot's action space. Real-world deployment requires a closed-loop policy that operates in the robot action space, which is why we use BC to learn robot actions from the data collected by the teacher, rather than using online distillation to learn from the teacher's action output.
>
> We also note that using two-stage RL + BC is a common approach in prior works [1,2,3] that transfer policies learned in simulation to the real robot.
>
> [1] Wang et al., UniGraspTransformer: Simplified policy distillation for scalable dexterous robotic grasping, CVPR 2025
> [2] Yin et al., Foundation Controller for Unprecedented Dexterity, 2025
> [3] Liu et al., DexTrack: Towards Generalizable Neural Tracking Control for Dexterous Manipulation from Human References, ICLR 2025
>
> **Q3.** How to align the robot controller between simulation and real and avoid collision damage.
>
> **A3.** For robot control, we use a PD controller and clip the difference between consecutive PD targets to ensure that the robot’s pose smoothly **follows the PD target with small tracking error**. For the Inspire Hand, we set a large stiffness in simulation to better align with the position controller used on the hardware. As a result, the controller gap between simulation and the real robot is small: replaying the same action sequence produces nearly the same motion trajectory in both settings. In addition, our demonstration-editing formulation **ensures that the RL policy produces smooth, continuous action sequences** similar to the demonstration, reducing dynamic or jerky motions that could exacerbate the sim-to-real gap. This allows sim-to-real transfer without additional domain randomization on PD controller parameters.
>
> We include a binary collision penalty in the reward function, which encourages the policy to minimize robot–table collisions. Appendix D.4 shows that **the collision probability converges to a very small value** during RL training. Even when contact with the table is necessary for grasping thin objects, **the robot arm's PD controller ensures that contact forces do not increase abruptly**, preventing damage to the Inspire Hand.
>
> **Q4.** Potential extension beyond the grasping task.
>
> **A4.** Extending our pipeline to more robotic manipulation tasks is a natural next step. DexMimicGen [4] has shown that replay-based methods can be applied to long-horizon bimanual manipulation tasks by dividing the task into multiple stages (e.g., pour water: grasp the right cup, pour into the left cup, place the right cup) and replaying the demonstration for each stage. This method can synthesize successful trajectories under varied object positions given one demonstration, but the object must remain the same.
> In future work, our framework can be applied to such tasks to generalize to novel objects via demonstration-editing RL. The basic idea is to train a multi-step RL policy that sequentially edits the demonstration for each stage and replays it, maximizing the task reward.
>
> [4] Jiang et al., DexMimicGen: Automated Data Generation for Bimanual Dexterous Manipulation via Imitation Learning, ICRA 2025

---

> > ### Author Response · Authors · 2025-11-23
> >
> > **Q5.** Randomization range for the object initial pose and its effect on RL.
> >
> > **A5.** We initialize the object within a 50 cm × 50 cm region (which almost covers the robot arm’s valid workspace) with a random orientation. This large initialization range does not harm RL performance, because in our framework, the edited demonstration is replayed in an object-centric manner. Thus, the optimal policy’s action output is invariant to the object’s position.

---

### Official Review · Reviewer_uPY2 · 2025-11-01

**Soundness:** 3
**Presentation:** 3
**Contribution:** 3
**Rating:** 8
**Confidence:** 4

**Summary:**

This paper presents DemoGrasp, a framework that first augments demonstrations with pose and object randomization and a reinforcement learning (RL) residual policy, and then distills the results into an imitation learning policy. Extensive experiments are conducted in both simulation and the real world. The ablation studies on RL-sampling comparison, camera configuration, and RL action spaces are well designed.

**Strengths:**

The experiments are extensive, and the real-world demonstrations looks robust.
The analysis of RL-sampling comparison, camera configuration, and action spaces is insightful.
RL residual learning and RGB sim-to-real transfer are challenging, and the paper addresses them successfully.
The presentation of the paper is clear and well structured.

**Weaknesses:**

- One weakness is that more details about the experimental settings should be provided, like RGB randomization details in the paper.
- It is suggested to replace some edited videos on the website with uncut, continuous grasping videos.

**Questions:**

- It would be helpful to discuss more recent related works in the paper.
- The quality of figures and videos can be improved, preferably with higher resolution.
- It would be good to include an open-source commitment in the paper to increase its reproducibility.

---

> ### Author Response · Authors · 2025-11-23
> **Thanks for your review! Here, we respond to your comments and address the issues. We hope to hear back from you if you have further questions!**
>
> **Q1.** Provide more details about the experimental settings and uncut videos.
>
> **A1.** We thank the reviewer for the valuable suggestions. We have added more implementation details regarding visual sim-to-real in Appendix E.3 in the revision. On our website, the video in the top-right corner shows 9 uncut recordings of our model repeatedly grasping and removing objects in cluttered scenes. Additionally, we have uploaded two 10-minute uncut videos, which demonstrate continuous grasping of 30 individual objects (highlighted with blue captions).
>
> **Q2.** Discuss more recent related works.
>
> **A2.** Related works on learning universal dexterous grasping policies are comprehensively discussed in Appendix B. We have revised this section to provide a deeper analysis of recent works, including UniGraspTransformer, RobustDexGrasp, ClutterDexGrasp, and DexGraspVLA. In this comparison, we highlight the contributions of our work, particularly its simplicity, effectiveness, extensibility to new embodiments, and high sim-to-real performance using only simple RGB images.
>
> **Q3.** Improve the resolution of figures and videos.
>
> **A3.**  Currently, we use a RealSense camera with a 640x480 resolution to record experimental videos. If the videos on the website appear fuzzy, it is likely because the website automatically switches to a 360p resolution. Please adjust the settings to switch to a higher resolution or download the videos directly. The newly uploaded uncut videos have higher resolution, and we plan to update the figures in the Appendix to higher resolution in the final revision.
>
> **Q4.** Open-source commitment.
>
> **A4.** We plan to open-source the code upon paper publication. This commitment has been added to Appendix H.

---

### Meta-Review · Area_Chair_Ywog · 2026-01-05

**Summary:**

This paper provides a way to generate diverse grasping data from a single demonstration. The key innovation is to treat the problem as one of demonstration editing where RL edits a grasp in a 1 step problem to solve grasping of new objects/scenarios. The results are strong and applied to a real robot and in simulation. Importantly they do with this just a simple success reward and a collision penalty. And the resulting "universal policy" seems to grasp lots of objects and generalize.

**Reviewer Concerns:**

Concerns raised were:
1. More details needed.
2. Some questions about the particular choice of sim2real distillation method.
3. Transferability of the method to totally new grasps was questioned.
4. Generality to more dynamic, dexterous manipulation tasks was questioned.
5. Some more questions on clarity.

Overall the reviewers answered most questions but said that other tasks beyond grasping is for future work.

**Reviewer Scores:**

I think most reviewers will leave their score unchanged, but kYw5 may be go up by 1, since the authors addressed all their concerns about transferability/compute in the response. I I think the 2 8s will likely not go higher.

---

### Decision · Program_Chairs · 2026-01-26

Accept (Poster)